# Unveiling the Secretome of the Fungal Plant Pathogen *Neofusicoccum parvum* Induced by *In Vitro* Host Mimicry

**DOI:** 10.3390/jof8090971

**Published:** 2022-09-17

**Authors:** Forough Nazar Pour, Bruna Pedrosa, Micaela Oliveira, Cátia Fidalgo, Bart Devreese, Gonzalez Van Driessche, Carina Félix, Nuno Rosa, Artur Alves, Ana Sofia Duarte, Ana Cristina Esteves

**Affiliations:** 1CESAM, Department of Biology, Campus Universitário de Santiago, University of Aveiro, 3810-193 Aveiro, Portugal; 2Department of Biochemistry and Microbiology, Laboratory of Microbiology, Ghent University, 9000 Ghent, Belgium; 3Faculty of Dental Medicine, Center for Interdisciplinary Research in Health (CIIS), Universidade Católica Portuguesa, 3504-505 Viseu, Portugal

**Keywords:** Botryosphaeriaceae, *Neofusicoccum parvum*, plant fungal interaction, secretome, LC-MS, *Eucalyptus globulus*

## Abstract

*Neofusicoccum parvum* is a fungal plant pathogen of a wide range of hosts but knowledge about the virulence factors of *N. parvum* and host–pathogen interactions is rather limited. The molecules involved in the interaction between *N. parvum* and *Eucalyptus* are mostly unknown, so we used a multi-omics approach to understand pathogen–host interactions. We present the first comprehensive characterization of the *in vitro* secretome of *N. parvum* and a prediction of protein–protein interactions using a dry-lab non-targeted interactomics strategy. We used LC-MS to identify *N. parvum* protein profiles, resulting in the identification of over 400 proteins, from which 117 had a different abundance in the presence of the *Eucalyptus* stem. Most of the more abundant proteins under host mimicry are involved in plant cell wall degradation (targeting pectin and hemicellulose) consistent with pathogen growth on a plant host. Other proteins identified are involved in adhesion to host tissues, penetration, pathogenesis, or reactive oxygen species generation, involving ribonuclease/ribotoxin domains, putative ricin B lectins, and necrosis elicitors. The overexpression of chitosan synthesis proteins during interaction with the *Eucalyptus* stem reinforces the hypothesis of an infection strategy involving pathogen masking to avoid host defenses. *Neofusicoccum parvum* has the molecular apparatus to colonize the host but also actively feed on its living cells and induce necrosis suggesting that this species has a hemibiotrophic lifestyle.

## 1. Introduction

*Eucalyptus* species are native to Australia but due to their enormous economic significance are planted in many countries around the world. *Eucalyptus* species were introduced in Portugal more than 100 years ago and are nowadays the most representative forest tree species. *Eucalyptus globulus* is the most abundant species in Portugal, occupying ca. 8500 km^2^, the equivalent to ca. 9% of the country (26% of the forest area of Portugal), mostly in Central and Northwest Portugal [1,2]. This species is well adapted to the Mediterranean-like climate and is exploited mainly due to the commercial interests of the pulp and paper industries. Unfortunately, they are commonly susceptible to diseases/infections caused by various species of the family Botryosphaeriaceae.

Botryosphaeriaceae are well-known fungal opportunistic pathogens that elicit disease symptoms in plants under stress conditions, resulting in high economic losses [3,4,5]. In addition, these species are known to occur in asymptomatic plant tissues as commensals or latent pathogens in a variety of tree species including *Eucalyptus* [5,6,7,8]. Botryosphaeriaceae have been associated with *Eucalyptus* canker and dieback in Portugal [9,10] and are considered a significant threat to the productivity and sustainability of *Eucalyptus* spp. plantations. In a survey conducted in 2015, and again in 2018, the predominant isolates collected from *Eucalyptus* were identified as belonging to the genus *Neofusicoccum* [9,10]. Several studies have reported a diverse assemblage of *Neofusicoccum* species occurring on *Eucalyptus* spp. both as disease-causing agents and as commensals [6,11,12].

*Neofusicoccum parvum* is a vascular aggressive pathogen that causes severe decline and dieback symptoms in a wide range of hosts [7,13], being also common in many *Eucalyptus* species [4,6,9,14,15]. In general, the fungus penetrates through wounds and colonizes the host tissues, causing shoot dieback, stem canker, cane bleaching, bud necrosis, and graft failure. *Neofusicoccum parvum* is an endophyte (i.e., it colonizes the interior of plants) that switches from a ‘no- or not visible’ inducing host damage status to a clear pathogen. In fact, not much is known about the strategies that this fungus employs to infect its hosts, or about the molecules it expresses during infection. Several studies have suggested that *N. parvum* pathogenicity could be related to the ability of this fungus to colonize woody tissue combined with the production of several phytotoxins [16,17,18,19] and also the expression of extracellular proteins with phytotoxic properties [20]. A study of genes encoding necrosis and ethylene-inducing proteins (NLPs) in *N. parvum* showed that they are functional genes encoding proteins toxic both to plant and mammalian cells, being most probably involved in virulence or cell death during *N. parvum* infection [21]. Recent genomic and transcriptomic analyses have shown that this pathogen has evolved special adaptive mechanisms to infect woody plants [22,23]. These mechanisms include a significant expansion of gene families associated with virulence and nutrient uptake, including cellular transporters, cell-wall-degrading enzymes (CWDEs), cytochrome P450s, putative effectors, and biosynthesis of secondary metabolites. The interaction between grapevine and *N. parvum* was also studied at the transcriptomic level [22,24]. Host plant stems and leaves underwent extensive transcriptomic reprogramming, but woody stems reacted earlier than leaves to infection. Gene expression analysis showed that *N. parvum* co-expresses genes associated with secondary metabolism and plant cell wall degradation in a dependent manner on the growth substrate and the stage of plant infection. Overall, these studies have shed light on the interactions between plants and *N. parvum*. However, a full understanding of the pathogenicity mechanism is still far from being accomplished. To investigate the mechanisms of pathogenicity of this fungus, we centered our analysis on the secretome [25,26,27,28,29], due to its relevance to the infection mechanism and to fungus–plant interactions. Proteomics data from the species of the family Botryosphaeriaceae are limited. So far, the proteome of *Diplodia seriata* [30], *Diplodia corticola* [31], and, most recently, *Lasiodiplodia theobromae* [32,33,34] have been made available. Proteins identified in these studies suggest differences in the infection strategies of these fungi. Although the genome of *N. parvum* was sequenced and released in 2013 [13], no proteomics studies have been carried out until now.

The aim of this study was to characterize the secretome of *N. parvum*, evaluate its response to an *in vitro* host mimicry, and predict interactions of the secretome proteins with host proteins.

## 2. Materials and Methods

### 2.1. Fungal Strains, Plant Material, and Culture Conditions

The strain used in this study, *N. parvum* CAA704, was recovered from *E. globulus* displaying symptoms of dieback and decline in Portugal [9]. This strain also proved to be pathogenic to *E. globulus* in artificial inoculation trials [9]. The strain was grown on Potato Dextrose Agar (PDA, Merck, Germany) at 25 °C for 7 days prior to the inoculations. The 3-months-old *E. globulus* (MB43, obtained from Altri, SGPS, S.A.) seedlings were watered weekly and kept at room temperature under natural light.

Two conditions were tested: control and infection-like. For the control condition, two mycelium plugs (5 mm diameter) were inoculated into a 250 mL flask containing 50 mL of Potato Dextrose Broth (PDB, Merck, Germany) and incubated in triplicate at 25 °C for 12 days. For the infection-like condition, a sterilized piece of *E. globulus* stem (±2 g) was added to the PDB, as described elsewhere [31]. The culture supernatant of each condition was harvested through filter paper and immediately stored at −80 °C for extracellular protein extraction. Mycelia obtained from both conditions were collected by filtration, washed with sterile water, and frozen with N_2_(L) for DNA and RNA extraction.

### 2.2. RNA Extraction and cDNA Synthesis

Total RNA was extracted from 12-days-old mycelium ground in liquid nitrogen (three biological replicates from each condition) using the Spectrum Plant Total RNA kit (Sigma-Aldrich, St. Louis, MO, USA), according to the manufacturer’s instructions. Samples were treated with DNase I digestion set (RNase-Free DNase Set, QIAGEN, Hilden, Germany) for 15 min to remove genomic DNA. The quality and quantity of RNA were checked by gel electrophoresis and NanoDrop™ 1000 Spectrophotometer (Thermo Scientific, Waltham, MA, USA). cDNA was generated using the Nzy First-Strand cDNA Synthesis Kit (Nzytech, Lisboa, Portugal), according to the manufacturer’s instructions.

### 2.3. Quantitative PCR

Target genes were selected according to their pattern of expression and functional annotation (Table 1). All reactions were performed in a CFX96 Real-Time thermocycler (BioRad, Hercules, CA, USA) using the NzySpeedy quantitative PCR (qPCR) Green Master Mix (2×) (NZYtech, Lisboa, Portugal). For each reaction, 5 µL of the Master Mix, 0.5 µL (10 µM) of each primer, 4.2 µL of nuclease-free water, and 0.5 µL of template cDNA were used. The PCR program used was: 95 °C—3 min, 40 cycles of 95 °C—15 s, and 60 °C—30 s. After this step, the fluorescence intensity was measured and, at the end of the program, the temperature was increased from 65 °C to 95 °C at a rate ramp of 0.1 °C/s, allowing the melting curves elaboration. Cq values were calculated with BIO-RAD CFX Manager software and used to compare the expression between reference and target genes.

### 2.4. Extracellular Protein Extraction

Secreted proteins were extracted using TCA/acetone according to the method described by Fernandes, et al. [31]. To discard precipitated polysaccharides, 35 mL of the culture supernatant was centrifuged (48,400× *g*, 1 h at 4 °C). One volume of cold TCA/acetone (20%/80% (*w*/*v*)) supplemented with 0.14% (*w*/*v*) dithiothreitol (DTT) was added to the supernatant and incubated at −20 °C for 1 h. Precipitated proteins were collected by centrifugation (15,000× *g*, 20 min, 4 °C) and the supernatant was removed. Precipitated proteins were washed with 10 mL of ice-cold acetone (twice) (15,000× *g*, 15 min, 4 °C) and 10 mL of ice-cold 80% acetone (*v*/*v*) (15,000× *g*, 15 min, 4 °C) to discard the excess of TCA from the precipitate. Residual acetone was air-dried, and the protein pellet was resuspended in 0.1 M Tris HCl pH 8 and stored at −80 °C.

### 2.5. Protein Sample Cleaning

To remove salts, detergents, and phenolic compounds, the protein extract was cleaned with the water/chloroform/methanol protein precipitation method (adapted from [36]). Briefly, a mixture of methanol, chloroform, and water (4:1:3 (*v*/*v*/*v*)) was added to the sample and thoroughly vortexed. Then, the mixture was centrifuged at 14,000× *g* for 1 min and the top aqueous methanol layer was removed (the proteins being in the interphase). Four volumes of methanol were added, and the mixture was vortexed and centrifuged at 14,000× *g* for 5 min. The supernatant was removed without disturbing the pellet. The air-dried pellet was finally resuspended in 0.1 M Tris HCl pH 8 and stored at −80 °C.

### 2.6. Protein Quantification

Protein concentration assay was carried out with the Pierce^®^ 660 nm Protein Assay kit (Thermo Scientific, Waltham, MA, USA) according to the manufacturer’s instructions, using Bovine Serum Albumin (BSA) as standard. All samples were quantified in triplicate.

### 2.7. Protein Quality Evaluation by Electrophoresis

The quality of protein samples was assessed by SDS–PAGE. Briefly, 3 µg of protein were denaturated and separated in a 12.5% SDS-PAGE gel electrophoresis, for 120 min at 120 V, in a Mini-PROTEAN 3 Cell (Bio-Rad, Hercules, CA, USA), according to Laemmli’s protocol [37]. The running buffer contained 100 mM Tris, 100 mM Bicine, and 0.1% (*w*/*v*) SDS. Gels were stained with Coomassie Brilliant Blue G-250. After staining, gels were scanned on a GS-800 Calibrated Densitometer (Bio-Rad, Hercules, CA, USA).

### 2.8. Tryptic Digestion, Mass Spectrometry Analysis, and Protein Identification

Ten μg of the protein sample were diluted in NH_4_HCO_3_ 50 mM buffer (in 30 μL). Twenty μL of BSA 0.002 μg/mL was added and the solution was incubated at 80 °C for 10 min. Samples were reduced with 5 μL of DTT 50 mM/NH_4_HCO_3_ 50 mM (incubation at 60 °C for 10 min) and alkylated with 5 μL of iodoacetamide (IAA) 150 mM/NH_4_HCO_3_ 50 mM (incubation in the dark for 20 min). Proteins were digested with 2 μL of trypsin 0.1 μg/μL. Afterward, samples were acidified with 1% formic acid and incubated at 37 °C for 30 min. After centrifugation (16,000× *g*, 30 min), the supernatant was transferred to new vials and a peptide purification step was performed using C_18_ Omix tips. The peptides were dried in a vacuum concentrator (SpeedVac, ThermoFisher Scientific, Waltham, MA, USA) and stored at −20 °C until analysis.

Purified peptides were re-dissolved in loading solvent (0.1% trifluoroacetic acid (TFA) in water/acetonitrile (ACN) (98:2, *v*/*v*)) and injected into an Ultimate 3000 RSLC nano system in-line connected to a Q Exactive HF mass spectrometer (Thermo, Waltham, MA, USA). Trapping was performed at 10 μL/min for 4 min in loading solvent A on a 20 mm trapping column (made in-house, 100 μm internal diameter (I.D.), 5 μm beads, C_18_ Reprosil-HD, Dr. Maisch, Ammerbuch, Germany) and the sample was loaded onto a 400 mm analytical column (made in-house, 75 µm I.D., 1.9 µm beads, C_18_ Reprosil-HD, Dr. Maisch, Ammerbuch, Germany). Peptides were eluted by a non-linear gradient from 2 to 56% solvent B [0.1% formic acid in water/acetonitrile (2:8, *v*/*v*)] over 145 min at a constant flow rate of 250 nL/min, followed by a 10 min wash reaching 97% MS solvent B and re-equilibration with solvent A (0.1% formic acid in water) for 20 min. The column temperature was kept constant at 50 °C by a column oven (Sonation COControl). The mass spectrometer was operated in data-dependent mode, automatically switching between MS and MS/MS acquisition for the 16 most abundant ion peaks per MS spectrum. Full-scan MS spectra (375–1500 *m*/*z*) were acquired at a resolution of 60,000 in the Orbitrap analyzer after accumulation to a target value of 3 × 10^6^. The 16 most intense ions above a threshold value of 1.3 × 10^4^ were isolated for fragmentation at a normalized collision energy of 28% after filling the trap at a target value of 1 × 10^5^ for a maximum of 80 ms. MS/MS spectra (200–2000 *m*/*z*) were acquired at a resolution of 15,000 in the Orbitrap analyzer.

The raw data generated from LC-MS was further inputted in Max-Quant (version 1.6.2.1, https://maxquant.net/maxquant/, accessed on 1 April 2017, Max Planck Institute, Martinsried, Germany), a quantitative proteomics software developed by Cox and Mann [38]. MS1 spectra were searched with the Andromeda peptide database engine [39] against a FASTA database of proteins from the *N. parvum* genome from UniProt (July, 2017) [40] and analyzed for label-free quantification of the peptides present in the samples. The peptide database was constructed from *in-silico* prediction of tryptic peptides with up to two missed cleavages, carbamidomethylation of cysteines as fixed modifications, and oxidation of methionines and N-terminal acetylation as variable modifications. Peptide spectral matches were validated using a percolator based on q-values at a 1% false discovery rate (FDR). Identified peptides were assembled into protein groups according to the law of parsimony and filtered to 1% FDR. Perseus software (version 1.6.1.3, https://maxquant.net/perseus/, accessed on 1 April 2017, Max Planck Institute, Martinsried, Germany) [41] enabled the affiliation of the protein groups into identified proteins. Identified proteins were filtered and only considered for analysis if present in 3 replicates and using at least 3 peptides for identification. Reverse proteins and proteins identified only by site were filtered out. A multi-scatter plot and hierarchical clustering were performed to assess the quality of the experiment. To identify interactor proteins, a two-sample *t*-test between control and infection-like samples was performed with minimal fold change (s0) of 1.8 and 1% FDR. A scatter plot, volcano plot, and profile plot were used to visualize the results (Appendix A).

### 2.9. Bioinformatic Analysis

Identified proteins were classified according to the GO (biological process). Whenever necessary, the protein’s family and domain were determined by the identification of conserved domains in the InterPro database (http://www.ebi.ac.uk/interpro, accessed on 1 April 2017) [42]. Cell-wall-degrading enzymes were classified according to the carbohydrate-active enzymes database CAZy (http://www.cazy.org, accessed on 1 April 2017) [43].

All proteins were analyzed for subcellular localization using the BaCelLo fungi-specific predictor [44], SignalP v4.1 (https://services.healthtech.dtu.dk/service.php?SignalP, accessed on 1 April 2017) [45], and SecretomeP predictor [46].

### 2.10. Interactomics Analysis

The OralInt algorithm [47] was used to predict the interactions between all the proteins of *Eucalyptus grandis* reference proteome (Uniprot UP000030711, 44,150 proteins) with the differentially secreted proteins of *N. parvum* identified in this study (117 proteins). OralInt is a computational prediction method based on an ensemble methodology combining five distinct protein–protein interactions (PPI) prediction techniques, namely: literature mining, primary protein sequences, orthologous profiles, biological process similarity, and domain interactions [47]. Since the sequence is the feature with the best overall performance, this method of predicting interactions can be applied independently of the organisms under study.

Interactions with a score ≥ 0.900 are represented using yFiles Organic Layout with Cytoscape 3.7.2 or in an edge bundling structure built using R (v4.1.2) [48] with packages ggraph (v2.0.5) [49], igraph (v1.2.11) [50], and tidyverse (v1.3.1) [51].

## 3. Results

### 3.1. Secretome Analysis

Prior to LC-MS, protein extracts were analyzed for quality control by SDS-PAGE (Appendix A). The secretomes of *N. parvum* grown in the absence (control) and presence of a *Eucalyptus* stem (infection-like condition) were analyzed. In total, 471 proteins were identified in both control and infection-like secretomes, of which 131 proteins were significantly different in abundance between the two conditions (*t*-test, difference cutoff of 1.8). Most of these proteins are extracellular (Table 2 and Appendix A), except for 14 proteins predicted as intracellular proteins (10.7%, Appendix A).

Proteins were classified according to their gene ontology (GO) (Molecular Function), and into 10 protein families: CAZymes, hydrolases, proteases, oxidoreductases, lyases, protein–protein interaction, carbohydrate-binding proteins, RNA-binding proteins, and proteins with other functions and unknown functions (Figure 1).

Among differentially secreted proteins, 74.6% were more abundant in infection-like conditions, while 24.5% were more abundant in control conditions (Table 2, Figure 1). Among induced proteins in the presence of *Eucalyptus*, we identified mainly CAZy proteins (50 proteins), esterases (9 proteins), proteases (4 proteins), oxidoreductases (5 proteins), and proteins with lyase activity (4 proteins) (Figure 1, Table 2).

Among the CAZy proteins, whose abundance is affected by the interaction with the *Eucalyptus* stem, glycosyl hydrolases (GH) are the most abundant group (68% of CAZymes), followed by proteins with auxiliary activities (AAs, 4 proteins), polysaccharide lyases (PLs, 8 proteins), carbohydrate esterases (CEs, 3 proteins) and unknown CAZy proteins (1 protein) (Table 2 and Appendix A). Esterases (EC 3.1.1.x) were more abundant in the infection-like conditions (Figure 1B).

A variety of proteases (endo and exoproteases) were also identified. The aspartic endopeptidase PEP1 (R1GM42) was more abundant in the infection-like condition (although its mRNA was as abundant as in control conditions, Appendix A) along with a variety of other metallopeptidases (M28, M35, and M43, Figure 1B and Table 2). In contrast, serine peptidases (S8 (R1EAW3) and S10 (R1FV38)) were less abundant in the infection-like conditions (Figure 1 and Table 2).

A putative berberine-like protein (R1GD68)—an oxidoreductase with a FAD-binding domain—was the most abundant protein in the infection-like condition (Table 2).

Other functional categories—proteins involved in carbohydrate binding (R1EYI5 and R1GAK8), RNA binding (R1ERG2, R1FZX2, and R1H1L9), protein–protein interactions (R1ENG6, R1E9S0, and R1GCJ5), and proteins with other functions (R1EGT1, R1GV87, R1EAF3, R1G1U2, R1FV21, R1FVG4, R1GBA7, R1EWZ5, R1GDV3, R1GKT0, R1E681)—were also identified (Figure 1 and Table 2).

### 3.2. Protein—Protein Interaction

Due to the lack of data available on the proteome sequence of *E. globulus*, the reference proteome for *E. grandis* (a closely related species) was used. Protein–protein interaction (PPI) networks between all the proteins of *E. grandis* (44,150 proteins, blue) and the extracellular proteins of *N. parvum* (117 proteins, red) were predicted using the OralInt algorithm (Figure 2). OralInt is based on high-quality experimental PPIs that feeds an artificial intelligence algorithm that is later validated. In previous studies, OralInt was applied to predict interactions between the Zika virus and the host [54] and between the oral microbiome and the host [55].

A total of 3201 interactions were predicted involving 76 proteins of *N. parvum* and 1591 proteins of *E. grandis*. Some proteins, both in *Eucalyptus* and in the fungus, showed a high number of interactions (Figure 2B, Table 3 and Appendix A). The functional analysis of *Eucalyptus* proteins on which the fungus acts is provided in the Appendix A. *Neofusicoccum parvum* hub proteins—those that center a high number of interactions—include mainly enzymes (Table 3). *Neofusicoccum parvum* proteins interact mainly with proteins involved in biosynthetic processes, nucleobase-containing compound metabolic processes, signal transduction, cell communication, response to endogenous stimulus, and response to stress, which fits well with a response to a foreign attack.

## 4. Discussion

Plant infection by phytopathogens, such as *N. parvum*, is a complex process that starts with the attachment of the infective propagule to the plant surface followed by penetration and infection. The infection mechanism of *Neofusicoccum* species relies on a myriad of molecules, mainly secondary metabolites, and proteins. It is known that species like *N. parvum* are able to express metabolites with phytotoxic activities such as Cyclohexenones, 5,6-Dihydro-2-pyrones, Melleins, Isosclerone, Hydroxypropyl- and methyl-salicylic acid, Tyrosol, Ethyl linoleate, Stearic acid, and Naphthalenones (Botryosphaerones A and D and 3,4,5-Trihydroxy-1-tetralone (for a review on the metabolites produced by *Neofusicoccum* spp. see [19]). These compounds were identified in isolates pathogenic of *Vitis vinifera*, and although it is expected that some of them will be present in other pathogen–host systems, it is not known. Understanding how other pathogen–host function is vital to fully understanding the mechanism of infection of *N. parvum*. The choice of the *Eucalyptus*-*N. parvum* system was based on the following reasons: (1) there are no reports on any molecules involved in the infection of *Eucalyptus* by *N. parvum* and (2) *Eucalyptus* is an economically vital crop in many countries, such as Portugal, and (3) there are no reports on the proteins involved in the infection mechanism of *N. parvum*.

*Neofusicoccum parvum* is a common pathogen of grapevine that infects many other hosts. In fact, the strain used in this work, *N. parvum* CAA704, was recovered from *E. globulus* displaying symptoms of dieback and decline and later it was shown to be pathogenic to *E. globulus* [9]. We compared the protein profiles of *N. parvum* in the control and infection-like conditions and identified and quantified proteins whose abundance changes in response to the *Eucalyptus* stem, to highlight proteins involved in the interaction with *Eucalyptus* during fungal infection. Although simple, and lacking the influence that molecules expressed by the host in response to the fungus attack have on the infection mechanism, the *in vitro* infection-like system used has been successfully used for other systems to mimic the infection mechanism of fungal pathogens [56,57].

Secreted proteins were visualized in a volcano plot (Figure 3) to have a quick visual identification of proteins that display large magnitude fold change and high statistical significance. The most significant proteins are discussed below.

Most of the proteins (86.3%) were predicted to contain a Signal P motif and are supposed to traverse the classical Golgi and endoplasmic reticulum secretion pathway. The possible implementation of the non-classical pathway for the proteins lacking signal peptide (13.7%) was confirmed using the SecretomeP predictor [46] (Table 2). Such proteins, known as leaderless-secreted proteins (LCPs), have been identified in several other studies involving the secretome [30,58,59]. Of these LCPs proteins, a putative ethanolamine utilization protein (R1G1U2) and a chitin-binding protein (R1EW80) showed low SecretomeP scores (NN score = 0.223 and 0.417, respectively) (Table 2). However, the NN score of a chitin-binding protein is relatively close to the 0.5 threshold, suggesting that the protein may in fact be secreted. The presence of intracellular proteins in the secretome is common and can result from cell death during culture, cell lysis during protein extraction, or secretion through non-common mechanisms. The number of cellular proteins identified in this study is similar to that identified in the secretome of *D. seriata* (16 proteins, [30]), *D. corticola* (12 proteins, [56]), and *L. theobromae* (16 proteins, [32]) the closest Botryosphaeriaceae species whose secretomes were studied.

### 4.1. Proteins Involved in Carbohydrate Metabolic Processes

Being a plant pathogen, it is not surprising that the *N. parvum* strain CAA704 secretome is mainly constituted by proteins involved in carbohydrate metabolic processes (18% and 30% of the proteins identified in control and in the presence of *Eucalyptus*, respectively) and in catabolic processes (8% and 9.7%; Figure 4). As a plant pathogen, *N. parvum* uses its host as a nutrient source and for that the carbohydrate-degrading enzymes are essential (Figure 4). Although there are very few secretomes of Botryosphaeriaceae fungi available, data shows that the trend is similar with a major component of the secretomes being carbohydrate-related enzymes [30,31,60,61]. Botryosphaeriaceae species can shift between a pathogenic and non-pathogenic lifestyle when triggered (by conditions not yet fully understood) and therefore the characterization of the secretomes under *controlled in vitro* conditions is of the utmost relevance to understanding the nature of these organisms.

Plant cell-wall-degrading enzymes (PCWDEs) play significant roles in plant colonization and are typical of necrotrophic life-style fungi [62], allowing them to perceive weak regions of plant epidermal cells and penetrate the plant’s primary cell wall. Our results indicate that *N. parvum* is equipped with an army of extracellular PCWDEs expressed even in the absence of plant tissue, but induced by the presence of the *Eucalyptus* stem (Table 2). Pectic enzymes (in multiple forms) are the first cell-wall-degrading enzymes induced by pathogens when cultured on isolated plant cell walls and the first produced in infected tissues [63,64]. Pectic enzymes induce the modification of the cell wall structure, exposing cell wall components for degradation by other enzymes [65]. Pectin is also present in *Eucalyptus* cell walls (15.2–25.8 mg g^−1^ pectin, [66]) and the secretion of pectin-degrading enzymes by *N. parvum* upon interaction with the *Eucalyptus* stem surely promotes the close interaction between the fungus and plant. All identified pectin-degrading enzymes [pectinases (GH53 and CE12) and pectate lyases (PL1, P3, PL4), Table 2 and Appendix A are more abundant in the presence of host material, suggesting that this fungus is more adapted to degrade intact or living plants than decaying biomass (where pectin is not present and is already decayed), which is in consonance with the fungus being a biotroph. Kang and Buchenauer [67] and Tomassini, et al. [68] demonstrated that wheat infection by *Fusarium culmorum* and *F. graminearum* depends on the production of CWDE at the early stages of infection which results in the facilitation of a rapid colonization of wheat spikes. Moreover, the up-regulation of these enzymes in lethal isolates of *Verticilium albo-atrum* compared to mild isolates was also described by Mandelc and Javornik [28], having implied its hypothetical contribution for plant vascular system colonization. We also identified cellulose-degrading enzymes mainly in the presence of the *Eucalyptus* stem. Putative GH12 protein (R1GQP5) raises special attention due to a high increase in response to *Eucalyptus* (3.9-fold up, Table 2). Recently, the xyloglucan-specific endo-β-1,4-glucanase (GH12 family) isolated from *P. sojae* culture filtrates induced cell death in dicot plants [69]. Gui, et al. [70] demonstrated that two of the six GH12 proteins in the fungus *Verticillium dahliae* Vd991 (VdEG1 and VdEG3) acted as virulence factors and as Pathogen-Associated Molecular Patterns (PAMPs), inducing cell death and triggering PAMP-triggered immunity in *Nicotiana benthamiana*. A glucanase (R1GZN3, GH7) was more abundant in the presence of the *Eucalyptus* stem than in control conditions. Cellulases belonging to GH6 and GH7 families are related to fungal virulence in the phytopathogenic fungus *Magnaporthe oryzae*, where they seem to be involved in the penetration of the host epidermis and further invasion [71].

Hemicellulases are generally involved in the degradation of hemicellulose from plant cell walls, helping in the colonization and in the acquisition of nutrients during infection. The up-regulation of two endoxylanases [beta-xylanase GH10 (R1FWZ0) and endo-1,4-beta-xylanase GH11 (R1GCT8)] was observed in *N. parvum* secretome in response to the *Eucalyptus* stem. GH10 and GH11 endoxylanases play significant roles in both vertical penetration of cell walls and horizontal expansion of the rice pathogen *M. oryzae* in infected leaves [72]. A recent study showed that two genes encoding GH10 xylanases are crucial for the virulence of the oomycete plant pathogen *Phytophthora parasitica* [73]. In *B. cinerea*, *Xyn11A* encodes an endo-β-1,4-xylanase Xyn11A, and the disruption of this gene resulted in reduced virulence of the pathogen [74]. However, several reports failed to show an essential role of endoxylanases in fungal pathogenicity [75,76,77]. Therefore, the role of xylanases in fungal pathogenesis may vary depending on the characteristics of the pathosystem and awaits further investigation.

The most abundant CAZy proteins in the control secretome of *N. parvum* are involved in lignin degradation (AA1, AA5, AA7) (Table 2 and Appendix A). These enzymes belong to the oxidoreductase family which can produce the H_2_O_2_ required for the action of extracellular peroxidases. Usually, *N. parvum* is not considered a major lignin-depolymerizing fungus, like white-rot fungi [78], but, in our study, several extracellular lignin-degrading enzymes were identified. However, most of those enzymes (five out of six) were less abundant in the presence of the *Eucalyptus* stem, indicating that they may have another role in *N. parvum* rather than a direct role in lignocellulose deconstruction during infection. A similar observation was described for the white-rot fungus *Lentinula edodes* when exposed to microcrystalline cellulose, cellulose with lignosulfonate, and glucose [79]. CAZymes involved in lignin degradation were repressed by cellulose. Cai, Gong, Liu, Hu, Chen, Yan, Zhou, and Bian [79] suggested that laccases may have a role in increasing fungal resistance to oxidative stress rather than being involved in lignocellulosic degradation.

### 4.2. Defense from Host

Oxidoreductases, important virulence factors induced during plant infection [80,81], are over-represented in the secretome of the *N. parvum* supplemented with the *Eucalyptus* stem (5-fold), where they may contribute to the alkaloid biosynthesis and production of hydrogen peroxide through the oxidation of metabolites [82].

Besides host degradation, fungal cell wall degradation plays a fundamental role in fungal development during infection, facilitating fungal branching and elongation [83]. The putative chitin-binding protein (R1EW80) contains a chitin deacetylase domain which catalyzes the conversion of chitin into chitosan required for appressorium formation [84]. Interestingly, the up-regulation of this protein in response to the host mimicry reinforces the hypothesis that this protein might play a role in host colonization. Some pathogens produce chitin-binding proteins that mask chitin, avoiding host recognition by shielding, or by modifying it [85,86]. Similarly, we identified a putative chitin deacetylase (CE4, R1E7G7) which is significantly up-regulated (5.6-fold) in the presence of the *Eucalyptus* stem. Chitin deacetylases are also involved in the protection of fungi from host plant chitinases by converting the fungal cell wall chitin into chitosan [87,88]. An endo-chitosanase (GH75, R1GTL6) is also up-regulated (4.1-fold) during the interaction with the *Eucalyptus* stem, suggesting that chitosan generation by chitin deacetylase enhanced the chitosanolytic activity of the fungus. We predicted that this endo-chitosanase (GH75, R1GTL6) has numerous interactions with other *Eucalyptus* proteins (69 proteins, Table 3 and Appendix A), some of which share important functions in *Eucalyptus* [auxin response factor (A0A059ACB3), delta-1-pyrroline-5-carboxylate synthase (A0A059BIT2), alpha-1,4 glucan phosphorylase (A0A059D8I8)], suggesting a central role for this protein in *N. parvum* infection mechanism of *Eucalyptus* (Table 3 and Appendix A).

### 4.3. Virulence

Most extracellular proteases identified—several of which were described as virulence factors in fungal necrotrophs [89,90,91,92,93]—were more abundant upon induction by *Eucalyptus*. Specifically, metalloproteases such as deuterolysin are also induced in a virulent strain of *D. corticola* upon challenge by the host (*Quercus suber*) [56]. It has been suggested that deuterolysin targets proteins in the plant cell wall [94], being directly involved in the infection mechanism.

The putative ricin B lectin protein (R1GAK8), involved in carbohydrate binding, contains a pectin_lyase_fold/virulence domain (InterPro IPR011050) considered a virulence factor in several species [95,96,97]. Ricin B lectins inhibit protein synthesis [98] and are highly expressed during infection [99,100]. In *N. parvum*, the putative ricin B lectin protein was induced in response to the host mimicry (4.3-fold).

Proteins containing ribonuclease/ribotoxin domains are also more abundant in the secretome of *N. parvum* supplemented with the *Eucalyptus* stem when compared to the axenic culture. Ribonucleases perform a variety of functions, serving as extra- or intracellular cytotoxins, and modulating host immune responses [101,102]. Ribotoxins are fungal extracellular ribonucleases that are highly toxic due to their ability to enter host cells and their effective ribonucleolytic activity against the ribosome [103]. Extracellular ribonucleases have been related to biotrophic fungi defenses, inhibiting the action of plant ribosome-inactivating proteins that would otherwise lead to host cell death, and pathogen death [102]. Secretion of low-molecular-weight guanyl-preferring ribonucleases (RNases) has also been reported in the secretome of the *D. corticola* [56]. Nonetheless, we predicted that the uncharacterized protein with ribonuclease activity (R1FZX2) establishes more than 360 interactions with *Eucalyptus* proteins.

*Neofusicoccum parvum* secretome contains necrotic elicitors like the necrosis-inducing protein (R1FZC0) and the putative epl1 protein (R1G1Q3) containing cerato-platanin domain, both in the control and infection-like conditions (Appendix A), suggesting that these phytotoxins are constitutively expressed by *N. parvum*. Interestingly, earlier we showed that NLP genes coding necrosis-inducing proteins in *N. parvum* are functional genes. NLP genes of *N. parvum* encode proteins toxic both to plant and mammalian cells, most probably involved in virulence or cell death during infection by *N. parvum* [21].

Proteins from the fasciclin family have been identified as cell adhesion molecules in various organisms [104,105,106,107]. In this study, the accumulation of the putative fasciclin domain family protein (R1EWZ5) in the infection-like conditions could be responsible for the attachment of fungal hyphae to the host material. In the rice blast fungus, *Magnaporthe oryzae*, MoFLP1 null mutants generated by targeted fasciclin gene disruption showed a significant reduction of conidiation, conidial adhesion, and appressorium turgor, resulting in overall decreased fungal pathogenicity [108]. But the knowledge on the role of fasciclin domain-containing proteins on fungi pathogenesis is still scarce and, according to Seifert [109], “cell adhesion” might be a result of turgor pressure and a buildup of adhesive materials such as pectin and not a direct function of fasciclin domain family proteins [109].

### 4.4. Protein–Protein Interactions

We predicted PPIs between *N. parvum* secreted proteins and proteins of *E. grandis* (used as a reference, since *E. globulus* genome sequence is not available). Some proteins identified display a high number of PPIs with host proteins, suggesting that those proteins may function as cross-talkers in biological functions between the fungus and the host. Among these, an auxin response factor (A0A059ACB3) in *Eucalyptus* raises special attention: not only it has a high number of interactions (33, Appendix A) but most proteins (27 out of 33 proteins), with whom the auxin response factor interacts, are more abundant in the infection-like condition. Furthermore, virulence factors like ricin B lectin protein (R1GAK8), putative exo-beta protein (PL3, R1H382), glucanase (R1GZN3, GH7), and putative pectate lyase (R1H2U7) were identified as the proteins interacting with auxin response factor (Appendix A). Auxin response factors (ARFs) family proteins are key players in auxin signaling [110]. Indole-3-acetic acid (IAA), the major form of auxin in plants, is the most important phytohormone with the main effects on plant growth, development, and on the regulation of plant senescence [111]. Pathogens may promote auxin accumulation or auxin signaling in the host through the action of virulence factors that have evolved to modulate host auxin biology. So far, studies on *Arabidopsis* imply that auxin reduces (hemi) biotroph resistance but enhances plant defenses toward necrotrophic pathogens [112]. Consistent with IAA promoting hemibiotroph susceptibility, auxin, and, more specifically, IAA, also act as virulence factors of the hemibiotrophic rice pathogens *Magnaporthe oryzae*, *Xanthomonas oryzae* pv. *oryzae*, and *X. oryzae* pv. *oryzicola* [113,114,115]. Like many other microorganisms, these pathogens produce and secrete IAA themselves and increase IAA biosynthesis and signaling upon infection [115].

## 5. Conclusions

Multi-omics approaches to pathogen–host interactions are becoming more common, but still largely rely on the existence of genome data. Nonetheless, even in the absence of such information, using similar genome data (e.g., from the same genus) has been widely used with success. We used proteomics (focused on the secretome) and computational interactomics to shed light on the infection mechanism of *N. parvum* on *Eucalyptus*. Most of the secretome proteins induced under host mimicry are, as in the case of other Botryosphaeriaceae fungi, cell-wall-degrading enzymes (CWDEs), especially those targeting pectin and hemicellulose, allowing the fungus to invade host tissues and extract nutrients for its own growth. Additionally, the degradation of xylan (hemicellulose) and pectin is required for fungal pathogens to invasively penetrate and proliferate inside host cells. We also found the up-regulation of chitosan synthesis and chitin degradation proteins during interaction with the *Eucalyptus* stem, suggesting that the pathogen masks itself to avoid plant defenses.

Surely not of negligible relevance, *N. parvum* proteins predicted to be involved in the largest number of interactions with *Eucalyptus* proteins are degradative enzymes. But besides attacking the integrity of the host, *N. parvum* appears to mask or modify its own cell surface avoiding plant defenses, which would allow the fungus to colonize the host while actively releasing enzymes and toxins (such as proteins containing ribonuclease/ribotoxin domains, putative ricin B lectins, putative epl1 proteins containing cerato-platanin domain and necrosis inducing proteins). The isolate used in this study is able to infect and cause disease in *E. globulus* [9]. Like many other species of Botryosphaeriaceae fungi, *N. parvum* can change between a commensal and a pathogenic lifestyle.

The distinction between an organism that colonizes living plants without causing symptoms of disease and that, given a certain stimulus, becomes a pathogen and an hemibiotroph (organisms that switch from an initial biotroph to necrotroph behavior) is not easy. *Neofusicoccum parvum* can colonize (infect?) its hosts without causing (visible) damages (compatible with being a non-pathogenic endophyte or—we believe this is a more accurate adjective—a commensal organism), but it can also become a pathogen, causing necrosis and ultimately the death of its host. The physiology of this shift is not known, although the literature refers that host stress can induce the shift.

It is our belief that the main issue is: is *N. parvum* a plant commensal that shifts to a pathogen? Or is it a pathogen that, for some time, does not express its pathogenic traits? Or is it a biotroph, feeding on its host, but without causing major harm?

What do we know for sure? We know that *N. parvum* can cause necrosis, so, at a given point of its life cycle, it is a necrotroph: it can express active necrosis elicitors able to induce phytotoxicity and necrosis [21]. The secretome of *N. parvum* in the presence of eucalyptus is also compatible with a necrotrophic lifestyle: it expresses CAZymes that are as important in establishing infection as in accessing nutrients during necrotrophic growth [116].

But is *N. parvum* a commensal or a (non-obligate) biotroph? The molecular evidence that we have indicates (“suggests”) that *N. parvum* is in fact a biotroph: it has the molecular machinery that allows it to colonize, spread, and feed on living plants (e.g., the overexpression of chitosan’s synthesis proteins). Furthermore, *N. parvum* expresses pathogenesis-related proteases and proteins containing ribonuclease/ribotoxin domains (known as toxic due to their ability to enter host cells and their effective ribonucleolytic activity against the ribosome) even in the absence of the plant host.

Last, the *N. parvum* genome contains some of the proteins involved in the shift between the biotroph and the necrotrophic phases of a typical hemibiotroph. One example is the 4-phosphopantetheinyl transferase protein (CgPPT1). CgPPT1 is functionally involved in and required for the biotrophy–necrotrophy transition of *Colletotrichum graminicola* [117]. In accordance with the hypothesis that *N. parvum* is an hemibiotroph, its genome contains a CgPPT1 gene. Also, in *Colletotrichum*, the shift from biotrophy to necrotrophy is defined by induction of the degradome, mirroring necrotrophic pathosystems [118]. *Neofusicoccum parvum* has the enzymes (CAZYmes, proteases, …) typical of these ‘degradomes’.

There are still many questions to be answered, but *N. parvum* secretome is more closely related to that of an hemibiotroph than to a plant commensal. It lives inside plants, feeds on them, and causes damage, eventually killing them. Why it remains “dormant” or why initial damages are not seen are questions that still need to be addressed.

## Figures and Tables

**Figure 1 jof-08-00971-f001:**
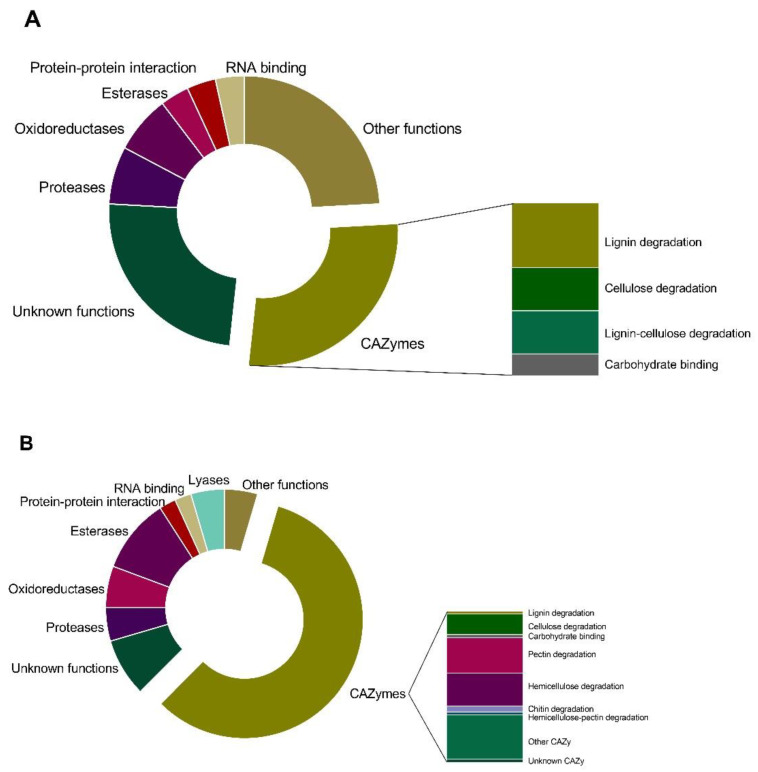
Functional classification (GO, Molecular Function) of the extracellular proteins secreted by *N. parvum* whose abundance was significantly different (*p* < 0.05) between the two conditions. (**A**) Proteins less abundant in the presence of the *Eucalyptus* stem, and (**B**) Proteins more abundant in the infection-like condition. For each category, the number of proteins is reflected by the size of the pie slice. The classification was obtained from the GO annotation at the UniProt database [40]. When lacking exact functional annotations in UniProt, the family and domain databases InterPro and Pfam [42,53] were used to reveal annotations of the identified proteins of conserved domains.

**Figure 2 jof-08-00971-f002:**
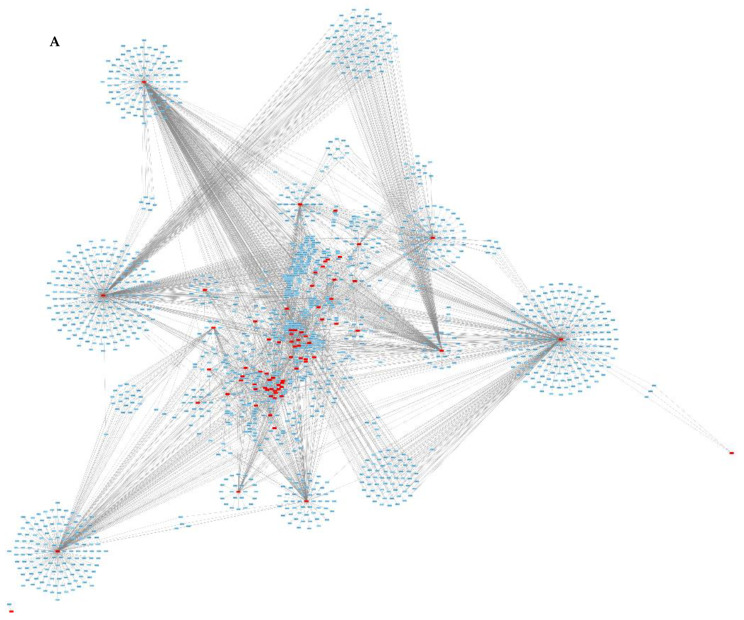
PPIs network prediction between secreted proteins from *N. parvum* (red) and the reference proteome of *Eucalyptus globulus* (blue). (**A**)—PPI interactions, the figure produced using Cytoscape v3.7.2. (**B**)—Visualization of the interactions between *N. parvum* proteins and the Biological Processes of the *Eucalyptus*-interacting proteins. The opacity of dots for each protein/protein category reflects the observed number of interactions.

**Figure 3 jof-08-00971-f003:**
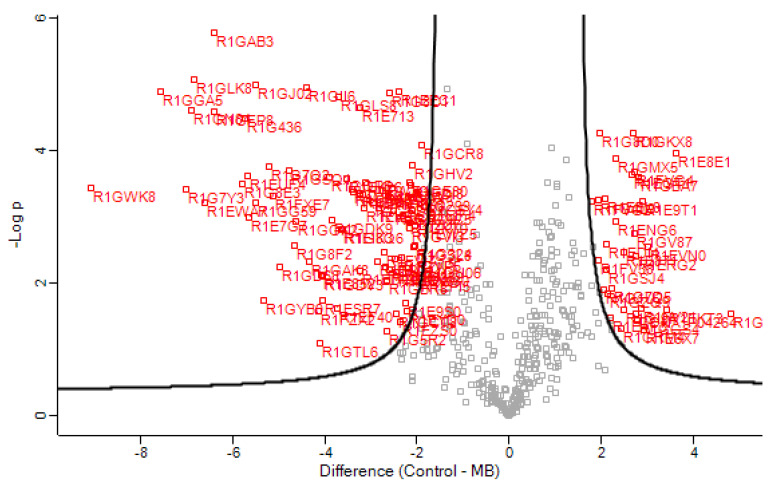
First volcano scatterplot of the samples under analysis. Proteins with a fold change <−2 or >2 are presented in red, and the remaining ones are in grey.

**Figure 4 jof-08-00971-f004:**
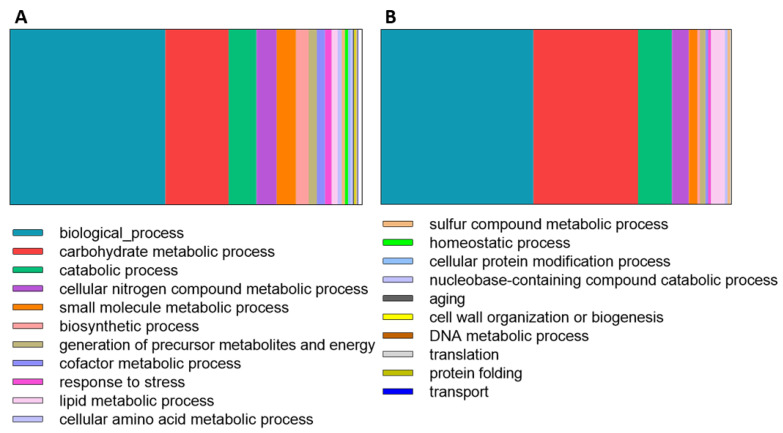
Gene Ontology (GO-Biological processes) of *Neofusicoccum parvum* proteins identified in (**A**)—control conditions and (**B**)—differentially expressed in the presence of the *Eucalyptus* stem.

**Table 1 jof-08-00971-t001:** Reference and target genes and respective primers.

Protein Name	Gene	Expression Condition	Primer Sequence (5′-3′)	Amplicon Length (bp)	Reference
Elongation factor 1-α	EF1α	Reference gene	FW: CGGTCACTTGATCTACAAGTGC RV: CCTCGAACTCACCAGTACCG	302	[35]
*Putative exo-beta protein* (PL3)	UCRNP2_317	Up-regulated	FW: ATTCAGCACTCCGGTACCAC RV: GCCGTCCACGGACTTGAT	255	Present study
*Putative aspartic endopeptidase PEP1 protein*	UCRNP2_6229	Up-regulated	FW: AGCTCCAGCTATGGTGGCTA RV: GACGATAGAGAAGCCGATGC	172	Present study

**Table 2 jof-08-00971-t002:** Summary of the proteins differentially secreted by *Neofusicoccum parvum* (CAA704). Protein localization was predicted using SignalP [52], SecretomeP 2.0 [46], and BaCeILO [44] tools.

Protein Name	Accession Number ^a^	Fold Change ^b^	*p*-Value ^c^	Unique Peptides ^d^	PEP ^e^	Intensity ^f^	Localization ^g,i^
**Cellulose degradation**							
GH5—Putative glycoside hydrolase family 5 protein	R1GZQ9	2.1	1.833	6	1.87 × 10^9^	169.3	Extracellular
GH5—Putative endoglucanase II protein	R1GLD6	1.9	1.554	6	8.42 × 10^8^	97.74	Extracellular
GH5—Putative cellulase family protein	R1G7G3	−2.7	3.403	12	2.36 × 10^10^	323.3	Extracellular
GH5—Putative endo-beta-protein	R1GDK9	−3.9	2.961	18	9.7 × 10^9^	323.3	Extracellular
GH3—Putative beta-d-glucoside glucohydrolase protein	R1EK26	−3.5	3.584	8	1.06 × 10^9^	98.06	Extracellular
GH3—Putative beta-glucosidase 1 protein	R1G324	−2	2.565	11	3.43 × 10^8^	84.3	Extracellular
GH7—Glucanase	R1GZN3	−2.5	2.374	10	1.49 × 10^10^	323.3	Extracellular
AA9/GH61/CBM1—Putative fungal cellulose-binding domain protein	R1GHV2	−2.1	3.779	7	2.05 × 10^9^	204.1	Extracellular
GH12—Putative glycoside hydrolase family 12 protein	R1GQP5	−3.9	3.605	8	1.05 × 10^10^	120.04	Extracellular
**Hemicellulose degradation**							
GH35—Putative beta-galactosidase B protein	R1E7W9	−2.5	3.430	21	4.23 × 10^9^	255.29	Extracellular
GH43—Putative glycosyl family protein	R1EP04	−2.9	2.330	6	5.29 × 10^8^	73.74	Extracellular
GH10—Beta-xylanase	R1FWZ0	−3.1	3.513	14	2.31 × 10^10^	323.31	Extracellular
GH43—Putative xylosidase: arabinofuranosidase protein	R1G299	−2.1	3.289	6	3.79 × 10^8^	140.27	Extracellular
GH43—Putative xylosidase glycosyl hydrolase protein	R1G5Y4	−1.9	3.227	13	3.51 × 10^10^	323.31	Extracellular
GH27—Alpha-galactosidase	R1G8C1	−2.6	4.883	12	1.8 × 10^10^	286.19	Extracellular NN ^h^ (0.861)
GH43—Putative galactan-beta-galactosidase protein	R1GG59	−5.5	3.216	14	3.67 × 10^9^	323.31	Extracellular
GH43—Arabinan endo-1,5-alpha-L-arabinosidase	R1GAB3	−6.4	5.780	10	4.97 × 10^9^	117.45	Extracellular
GH51—Putative alpha-l-arabinofuranosidase a protein	R1EVS4	−3.1	3.133	10	7.22 × 10^8^	190.73	Extracellular
CE5—Putative acetylxylan esterase protein	R1EWW2	−2.3	1.059	2	1.54 × 10^9^	323.31	Extracellular NN ^h^ (0.898)
GH11—Endo-1,4-beta-xylanase	R1GCT8	−2.4	1.534	7	3.41 × 10^8^	144.76	Extracellular
GH43/CBM6—Putative glycosyl hydrolase family 43 protein	R1GE80	−2.2	3.527	16	8.42 × 10^9^	307.72	Extracellular
**Lignin degradation**							
AA5—Putative glyoxal oxidase protein	R1EDI4	2.1	2.596	10	1.32 × 10^9^	86.505	Extracellular
AA1—Putative laccase-1 protein	R1G4L9	1.9	3.262	11	1.03 × 10^10^	227.45	Extracellular
AA7—Putative FAD-dependent oxidoreductase protein	R1FVT8	2.2	1.428	14	9.82 × 10^8^	123.3	Extracellular
AA3—Putative alcohol dehydrogenase protein	R1EH41	−2.7	2.253	3	2.58 × 10^8^	35.849	Extracellular NN ^h^ (0.648)
**Lignin/celulose degradation**							
AA3/CBM1—Putative cellobiose dehydrogenase protein	R1H3M7	2	1.856	16	1.72 × 10^9^	157.89	Extracellular
AA3—Putative GMC oxidoreductase protein	R1FVG2	1.8	3.233	23	5.79 × 10^10^	323.31	Extracellular NN ^h^ (0.655)
**Pectin degradation**							
GH53—Arabinogalactan endo-beta-1,4-galactanase	R1G7Y3	−7	3.424	9	6.29 × 10^9^	161.62	Extracellular
CE12—Putative rhamnogalacturonan acetylesterase protein	R1GFP8	−6.4	4.594	9	9.04 × 10^9^	155.37	Extracellular
GH53—Arabinogalactan endo-beta-1,4-galactanase	R1GVP5	−2.3	2.391	5	5.89 × 10^8^	56.599	Extracellular
PL3—Putative pectate lyase protein	R1EWA7	−6.6	3.224	9	6.54 × 10^9^	297.03	Extracellular
PL3—Putative pectate lyase protein	R1GN84	−6.2	4.605	6	4.57 × 10^9^	103.84	Extracellular
PL1—Putative pectate lyase a protein	R1GII6	−4.4	4.962	13	1.75 × 10^10^	323.31	Extracellular
PL4—Putative rhamnogalacturonan lyase protein	R1GJ02	−5.5	4.999	18	3.48 × 10^9^	227.89	Extracellular
PL1—Putative pectate protein	R1GSQ1	−4.8	3.712	5	1.4 × 10^9^	91.554	Extracellular NN ^h^ (0.592)
PL3—Putative exo-beta-protein	R1H382	−2.2	2.235	24	1.8 × 10^11^	323.31	Extracellular NN ^h^ (0.798)
GH28—Putative extracellular exo-protein	R1GW72	−3.2	3.359	5	8.03 × 10^8^	57.212	Extracellular
PL4—Rhamnogalacturonate lyase	R1EPI5	−2	2.077	6	4.03 × 10^8^	107.86	Extracellular
PL4—Rhamnogalacturonate lyase	R1GGA5	−7.6	4.898	23	1.38 × 10^10^	323.31	Extracellular
**Chitin degradation**							
CE4—Putative chitin deacetylase protein	R1E7G7	−5.6	2.993	6	3.73 × 10^9^	53.089	Extracellular
GH75—Endo-chitosanase	R1GTL6	−4.1	1.099	4	4.37 × 10^9^	59.996	Extracellular
**Other CAZY**							
GH16—Putative glycoside hydrolase family 16 protein	R1EVI7	−2.7	2.462	3	2.12 × 10^9^	34.095	Extracellular
**Esterase**							
Carboxylic ester hydrolase	R1GKX8	2.7	4.271	7	5.44 × 10^8^	70.895	Extracellular
Putative GDSL-like lipase acylhydrolase protein	R1E852	−4.1	2.117	6	3.13 × 10^9^	323.31	Extracellular
Carboxylic ester hydrolase	R1E8C5	−2.8	2.209	9	7.04 × 10^8^	79.717	Extracellular
Putative GDSL-like lipase acylhydrolase protein	R1GK66	−2.6	3.165	6	4.2 × 10^8^	40.614	Extracellular
Carboxylic ester hydrolase	R1GSL8	−2.1	2.550	6	1.91 × 10^8^	83.343	Extracellular
Putative carboxylesterase protein	R1EIK3	−3.7	1.530	4	7.19 × 10^8^	37.138	Extracellular NN ^h^ (0.768)
Carboxylic ester hydrolase	R1G8E3	−5.8	3.503	9	2.21 × 10^9^	171.03	Extracellular
Putative carboxylesterase family protein	R1G9C5	−2.1	3.136	5	1.71 × 10^8^	39.971	Extracellular
Putative GDSL lipase acylhydrolase family protein	R1EIF4	−1.8	2.818	6	3.13 × 10^9^	134.94	Extracellular NN ^h^ (0.756)
Carboxylic ester hydrolase/tannase family	R1GJW0	−1.9	3.095	20	4.01 × 10^9^	323.31	Extracellular
**Protease**							
Peptidase S1 family—putative carboxypeptidase S1 protein	R1FV38	1.9	2.351	7	4.06 × 10^9^	175.54	Extracellular
Peptidase S8 family—putative peptidase S8 S53 subtilisin kexin sedolisin protein	R1EAW3	2.2	1.289	5	1.54 × 10^9^	157.52	Extracellular
Peptidase A1 family—Putative aspartic endopeptidase PEP1 protein	R1GM42	−3.2	4.661	4	4.3 × 10^9^	98.628	Extracellular
Peptidase M43—Putative metalloprotease protein	R1FXE7	−5.1	3.311	5	3.6 × 10^9^	134.74	Extracellular
Peptidase M28 family—peptide hydrolase	R1GBR8	−2.7	2.039	6	1.35 × 10^9^	209.59	Extracellular
Peptidase M35 family—neutral protease 2	R1EL46	−2.3	0.943	5	1.62 × 10^9^	102.51	Extracellular
**Oxidoreductase**							
Putative FMN-dependent dehydrogenase protein	R1E6X7	2.9	1.290	16	6.56 × 10^8^	127.23	Extracellular
Putative FAD-binding domain-containing protein	R1E8E1	3.6	3.973	11	4.38 × 10^9^	264.43	Extracellular
Putative cyclohexanone monooxygenase protein	R1EF40	−3.7	1.628	2	9.32 × 10^9^	20.921	Extracellular
Putative tyrosinase central domain protein	R1ERX8	−2.4	2.164	9	8.84 × 10^8^	90.821	Extracellular NN ^h^ (0.817)
Putative FAD FMN-containing dehydrogenase protein	R1GB06	−3.4	3.369	16	9.58 × 10^8^	192.5	Extracellular
Putative berberine-like protein	R1GD68	−5	2.241	13	2.88 × 10^9^	323.31	Extracellular
Putative GMC protein	R1ELQ0	−2.1	0.517	6	5.13 × 10^9^	40.919	Extracellular
**Lyase**							
Putative pectate lyase protein	R1H2U7	−2.3	3.013	4	2.56 × 10^8^	28.73	Extracellular
Putative-secreted protein	R1GFS9	−3	3.218	20	2.59 × 10^11^	323.31	Extracellular
Putative pectate lyase protein	R1G436	−5.7	4.498	17	3.09 × 10^9^	217.8	Extracellular
Uncharacterized protein	R1GU06	−1.9	2.289	7	2.48 × 10^10^	323.31	Extracellular
**Protein–protein interaction**							
Putative six-bladed beta-propeller-like protein	R1ENG6	2.3	2.942	3	4.52 × 10^8^	36.528	Extracellular
Putative six-bladed beta-propeller-like protein	R1E9S0	−2.3	1.704	2	4.15 × 10^8^	21.736	Extracellular
Putative SMP-30 gluconolaconase LRE-like region protein	R1GCJ5	−2.1	0.903	5	1.13 × 10^9^	92.83	Extracellular NN ^h^ (0.754)
**Carbohydrate binding**							
Putative alpha-mannosidase family protein	R1EYI5	1.8	1.196	2	6.81 × 10^8^	23.805	Extracellular
Putative ricin B lectin protein	R1GAK8	−4.3	2.336	6	1.66 × 10^9^	97.147	Extracellular
**RNA binding**							
Putative ribonuclease T2 protein	R1ERG2	2.8	2.404	2	5.88 × 10^8^	62.528	Extracellular
Uncharacterized protein	R1FZX2	−4.1	1.575	6	1.9 × 10^9^	43.942	Extracellular
Putative extracellular guanyl-specific ribonuclease protein	R1H1L9	−2.1	0.586	3	4.24 × 10^9^	48.559	Extracellular
**Other function**							
Putative allergen V5 Tpx-1-related protein	R1EAF3	2.2	3.639	5	5.58 × 10^9^	181.15	Extracellular
Putative ethanolamine utilization protein	R1G1U2	2	2.760	5	4.1 × 10^8^	54.96	Extracellular NN ^h^ (0.223)
Putative ABC-type Fe^3+^ transport system protein	R1FV21	2.9	1.484	6	4.15 × 10^8^	58.409	Extracellular
Putative major royal jelly protein	R1FVG4	2.7	1.896	15	5.75 × 10^10^	323.31	Extracellular
Putative ABC-type Fe^3+^ transport system protein	R1GBA7	2.8	1.611	15	7.33 × 10^10^	323.31	Extracellular
Putative alpha beta hydrolase protein	R1EGT1	2.7	3.692	11	3.31 × 10^9^	118.1	Extracellular
Putative glutaminase protein	R1GV87	2.7	3.603	10	1.15 × 10^9^	215.65	Extracellular
Putative fasciclin domain family protein	R1EWZ5	−2	2.878	12	2.37 × 10^9^	129.86	Extracellular
Uncharacterized protein	R1GDV3	−4.1	2.108	5	3.67 × 10^10^	323.31	Extracellular
Putative BNR Asp-box repeat domain protein	R1GKT0	−2.2	2.950	11	2.45 × 10^10^	323.31	Extracellular
Putative extracellular aldonolactonase protein	R1E681	−1.8	0.892	5	2.15 × 10^9^	183.99	Extracellular
**Unknown**							
Putative extracellular serine-threonine rich protein	R1E9T1	2.9	3.242	3	8.48 × 10^8^	78.551	Extracellular
Putative membrane-spanning 4-domains subfamily a member 14 protein	R1EE60	2.9	3.242	3	8.48 × 10^8^	78.551	Extracellular
Uncharacterized protein	R1EBL8	2.1	3.282	11	1.26 × 10^10^	323.31	Extracellular
Putative GPI anchored cell wall protein	R1G7D5	2.2	1.914	4	1.35 × 10^9^	24.812	Extracellular
Uncharacterized protein	R1GMX5	2.3	3.894	6	2.11 × 10^10^	185.31	Extracellular
Uncharacterized protein	R1GRM4	2.4	1.314	4	7.62 × 10^8^	37.391	Extracellular
Uncharacterized protein	R1G5W7	2.1	0.681	2	8.78 × 10^8^	20.89	Extracellular
Uncharacterized protein	R1ESR7	−4	1.737	3	1.4 × 10^10^	48.44	Extracellular
Putative-secreted protein	R1G8U3	−3.4	3.421	6	4.85 × 10^8^	49.633	Extracellular
Uncharacterized protein	R1GYB0	−5.3	1.742	7	6.21 × 10^9^	132.23	Extracellular
Putative GPI anchored cell wall protein	R1ENT4	−2.4	0.948	4	1.24 × 10^9^	40.251	Extracellular
Putative 34-dihydroxy-2-butanone 4-phosphate synthase protein	R1EY60	−2.1	1.093	2	3.23 × 10^8^	44.267	Extracellular
Uncharacterized protein	R1GLY2	−2.1	1.562	6	3.69 × 10^8^	67.175	Extracellular
Putative exo-beta-glucanase protein	R1G5R2	−2.6	1.282	7	1.19 × 10^11^	323.31	Extracellular

^a^ Protein accession provided by the UniProtKB database [40]; ^b^ Fold change: the difference between the average intensities of two groups (log ratio control vs infection-like); Negative fold change values indicate proteins are more abundant in the infection-like secretome and positive fold change values indicate proteins are more abundant in the control secretome; ^c^
*p*-value: displaying significance which is expressed as -log values; ^d^ Unique peptides: The total number of unique peptides associated with the protein group (i.e., these peptides are not shared with another protein group); ^e^ PEP: Posterior Error Probability of the identification. This value essentially operates as a *p*-value, where smaller is more significant; ^f^ Intensity: Summed up extracted ion current (XIC) of all isotopic clusters associated with the peptide sequence, and protein intensities summed the intensities of all peptides assigned to the protein group; ^g^ Signal prediction calculated by using the SignalP [52]; ^h^ NN: Non-classically secreted proteins analyzed with SecretomeP 2.0 [46]; Proteins with NN score ≥ 0.5 were considered unconventionally secreted; ^i^ Protein localization was predicted by the BaCeILO predictor [44].

**Table 3 jof-08-00971-t003:** Summary of the proteins with the highest number of PPI between proteins differentially secreted by *Neofusicoccum parvum* (CAA704) and *Eucalyptus grandis* proteins. The “Degree” stands for the number of interactions.

Protein Name	Accession Number	Degree	Organism
Putative gmc protein	R1ELQ0	419	*Neofusicoccum parvum* (strain UCR-NP2)
Uncharacterized protein	R1G5W7	406	*Neofusicoccum parvum* (strain UCR-NP2)
Uncharacterized protein	R1FZX2	367	*Neofusicoccum parvum* (strain UCR-NP2)
Putative metalloprotease protein	R1FXE7	258	*Neofusicoccum parvum* (strain UCR-NP2)
Putative alpha-mannosidase family	R1EYI5	225	*Neofusicoccum parvum* (strain UCR-NP2)
Putative cyclohexanone monooxygenase	R1EF40	166	*Neofusicoccum parvum* (strain UCR-NP2)
Putative GDSL-like lipase acylhydrolase	R1GK66	154	*Neofusicoccum parvum* (strain UCR-NP2)
Putative alcohol dehydrogenase protein	R1EH41	117	*Neofusicoccum parvum* (strain UCR-NP2)
Endo-chitosanase	R1GTL6	69	*Neofusicoccum parvum* (strain UCR-NP2)
Uncharacterized protein	R1ESR7	69	*Neofusicoccum parvum* (strain UCR-NP2)
Auxin response factor	A0A059ACB3	33	*Eucalyptus grandis*
Histone H3	A0A059AF37	28	*Eucalyptus grandis*
Histone H3	A0A059BQE5	20	*Eucalyptus grandis*
Protein kinase domain-containing protein	A0A059CUY0	19	*Eucalyptus grandis*
HATPase_c domain-containing protein	A0A059DD44	17	*Eucalyptus grandis*
Glyco_transf_20 domain-containing protein	A0A059CZ70	17	*Eucalyptus grandis*
Uncharacterized protein	A0A059CUY2	17	*Eucalyptus grandis*
Protein kinase domain-containing protein	A0A059CBV7	16	*Eucalyptus grandis*
ERCC4 domain-containing protein	A0A059C0I5	16	*Eucalyptus grandis*
Na_H_Exchanger domain-containing protein	A0A059DJ06	15	*Eucalyptus grandis*

## Data Availability

The data presented in this study are available in Appendix A.

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
