# Peer review of "Unveiling the Secretome of the Fungal Plant Pathogen Neofusicoccum parvum Induced by In Vitro Host Mimicry"

_jof, 2022, doi:10.3390/jof8090971_

Round 1
Reviewer 1 Report
The manuscript by Nazar Pour and co-author reports the first comprehensive characterization of the in vitro secretome of N. parvum and a prediction of protein-protein interactions using a dry-lab non-targeted interactomics strategy. The manuscript is well written, and the degree of novelty is very high. Moreover, the results reported in the manuscript can be valuable for N. parvum's investigation of other important crops, such as grapevine.
Although, here are some suggestions to further improve the manuscript:
1) I suggest including information on phytotoxic metabolites produced by N. parvum and how the secretome proteins can interact to modulate plant defences, to further improve the discussion section. Moreover, I propose discussing some perspectives on the application of multi-omics analysis in planta studies.
2) I suggest moving the volcano plot Figure S5, to the main manuscript
3) I suggest replacing the reference number 16 with the more recent: Masi, M., Cimmino, A., Reveglia, P., Mugnai, L., Surico, G., & Evidente, A. (2018). Advances on fungal phytotoxins and their role in grapevine trunk diseases. Journal of Agricultural and Food Chemistry, 66(24), 5948-5958.
Reviewer 2 Report
REVIEW jof-1854367
The manuscript entitled “Unveiling the secretome of the fungal plant pathogen Neofusicoccum parvum induced by in vitro host mimicry“ by Pour et al. compares the secretomes of the Eucaliptus globulus pathogen Neofusicoccum parvum obtained during in vitro culture. Two conditions were assayed, the control (growth on PDF) and the infection-like (growth on PDF plus Eucaliptus stem). Besides this analysis, a prediction of the possible interactions between some of the extracellular proteins of N. parvum and proteins of E. grandis was performed by means of an algorythm.
This work is purely descriptive and is based on the assumption that in vitro growth of the fungal pathogen when Eucaliptus stem is added to the medium mimics host infection. The problem with this assumption is that Eucaliptus stem is not the real host but mainly food for the pathogen, therefore one may expect that many or most of the differentially expressed genes would be the ones related to degradation of the sugars present in the stem. Actually, this is the result obtained, as the most abundant proteins secreted in the “infection-like” condition are enzymes involved in cell wall degradation. The results obtained are not different from the ones obtained from several other similar analyses, which diminishes the originality of the work. The results support the already known general picture about host colonization by fungal pathogens and it does not provide new information on the E. globulus-N. parvum interaction.
The authors state in the introduction that Neofusicoccum species include both isolates able to infect and cause disease on Eucaliptus sp. and endophytes that colonize the host without causing disease. It would have been very interesting to compare the secretomes of pathogenic and endophytic strains with the aim of shedding light on the proteins specific to the pathogenic style of life. In the absence of this comparative study, the results obtained are of limited value as they could be specific of a pathogenic host colonization or an endophytic host colonization. The authors do not give any clue that might be indicative of one or the other (for instance, are there toxin-like proteins present in the secretome of the infection-like condition?).
Besides the identification of the proteins present in the secretome the authors also use the OralInt algorithm to predict host-pathogen protein interactions. This analysis, although interesting, has two drawbacks. First, the the host species is E. grandis and not E. globulus. The pathogenicity of the N. parvum isolate used in this study has been confirmed on E. globulus, but not in E. grandis. Therefore, it is not a verified host. Second, the use of a single algorithm makes this analysis tentative.
Despite the scarcity of experimentally confirmed results, beyond the in silico analysis, the authors dedicate more than 4 pages to a very long Discussion section. Most of it is a review of the roles of the main protein families described.
The conclusions shown at the end of the manuscript should be reviewed in depth. In the second paragraph the authors explain the absence in the secretome of lignin degrading enzymes on the basis of the inability of endophytes to degrade complex substrates as lignin. This assumption is not supported by references or experimental data. Furthermore, it arises a serious concern on the nature of the isolate used in the study: is it a pathogen or an endophyte? This issue should be clarified.
The third paragraph of the conclusions section presents as a result what is just indirect evidence obtained from a single in silico analysis. The whole paragraph should be rewritten and the tense changed to show that it is a working hypothesis.
The main question that arises after reading the manuscript is: are the results obtained evidence of pathogenicity (thus, specific to the pathogenic isolates of N. parvum) or only colonization (shared by both pathogenic and endophytic isolates)? What is the answer of the authors to this question?
Minor comments
Fig. 1: the colors used to depict the segments in the graphic corresponding to the different types of proteins are not the same in panel A and panel B, what makes it difficult to compare one to the other.
Table 2. It should be a supplementary figure, as the information that contains is not of much importance for the results. Furthermore, it contains redundant information, such as the localization of the proteins (extracellular for all of them, what should be the expectation in a secretome analysis). It could be substituted by a shorter table listing the most abundant proteins.
Round 2
Reviewer 2 Report
REVIEW jof-1854367 resubmission
The new version resubmitted by the authors includes several changes following the reviewers recommendations. In general the manuscript has been improved, but it is in need of some further modifications.
The Discussion section, although still very long to my taste, is now more readable as it has been distributed in several subsections. Even more important, the subsection on virulence has been strengthened with the addition of a new paragraph on the role of proteins of the fascidin family.
The conclusions section has also been deeply rewritten. The sentence about the expression of lignin degrading enzymes, which was very confusing, has been removed. Moreover, in the authors’ response to my criticisms they state a very important issue: the isolate analyzed has the molecular tools to behave either as a pathogen or an endophyte. Together with some other statements made by the authors in their response, a very important question has been answered, at least partially. In my former review I said that the main question that arises after reading the manuscript is: are the results obtained evidence of pathogenicity (thus, specific to the pathogenic isolates of N. parvum) or only colonization (shared by both pathogenic and endophytic isolates)? It is now quite clear to me that isolates of N. parvum may behave both as pathogens and endophytes. It is not the species that may include both pathogenic and endophytic isolates, as it happens in many plant pathogenic species, but the same isolate that may behave as one or the other. This is something that is not clearly stated in the manuscript. Therefore I recommend that a paragraph should be included in the Introduction section highlighting this issue, with references that give some backup.
Accordingly to this line of reasoning, I believe that the most important result of this work is the confirmation that the genome of N. parvum isolates contains genes that allow this organism to develop a pathogenic or an endophytic way of life and that the expression of virulence genes (required for the pathogenic way of life) is enhanced by in vitro host mimicry. This is clearly understood from the authors’ response but I believe is not as clearly stated in the manuscript. For instance, in the last phrase of the Conclusions section the authors say that the isolate analyzed has the molecular armament to both be an endophyte and also become a pathogen. I agree to that. But then, the sentence continues “suggesting that this species has an hemibiotrophic lifestyle”. This is confusing because it seems that the authors mistake the possibility of a certain isolate to behave as an endophyte or a pathogen (may be in response to external stimuli) with the fact that a hemibiotroph first invade the host living cells and then transitions to a necrothophic lifestyle to obtain nutrients by killing the host cells.
In my opinion, with the suggested modifications aimed to clarify these issues the manuscript shall be ready for publication.
Minor comments
Fig. 1: the colors used to depict the segments in the graphic corresponding to the different types of proteins are not the same in panel A and panel B, what makes it difficult to compare one to the other.
This has not been modified.
Author Response
Thank you so much for your suggestions. Please see the attachment.
